# GM1 Oligosaccharide Efficacy in Parkinson’s Disease: Protection against MPTP

**DOI:** 10.3390/biomedicines11051305

**Published:** 2023-04-28

**Authors:** Maria Fazzari, Giulia Lunghi, Alexandre Henriques, Noëlle Callizot, Maria Grazia Ciampa, Laura Mauri, Simona Prioni, Emma Veronica Carsana, Nicoletta Loberto, Massimo Aureli, Luigi Mari, Sandro Sonnino, Elena Chiricozzi, Erika Di Biase

**Affiliations:** 1Department of Medical Biotechnology and Translational Medicine, University of Milano, 20054 Segrate, MI, Italy; maria.fazzari@unimi.it (M.F.); giulia.lunghi@unimi.it (G.L.); maria.ciampa@unimi.it (M.G.C.); laura.mauri@unimi.it (L.M.); simona.prioni@unimi.it (S.P.); emma.carsana@unimi.it (E.V.C.); nicoletta.loberto@unimi.it (N.L.); massimo.aureli@unimi.it (M.A.); sandro.sonnino@unimi.it (S.S.); 2Neuro-Sys, 410 Chemin Départemental 60, 13120 Gardanne, France; alexandre.henriques@neuro-sys.com (A.H.); noelle.callizot@neuro-sys.com (N.C.); 3Department of Immunology, St. Jude Children’s Research Hospital, Memphis, TN 38105, USA; luigi.mari@stjude.org

**Keywords:** GM1 ganglioside, GM1 oligosaccharide, Parkinson’s disease, MPTP, neuroprotection, plasma membrane signaling

## Abstract

Past evidence has shown that the exogenous administration of GM1 ganglioside slowed neuronal death in preclinical models of Parkinson’s disease, a neurodegenerative disorder characterized by the progressive loss of dopamine-producing neurons: however, the physical and chemical properties of GM1 (i.e., amphiphilicity) limited its clinical application, as the crossing of the blood–brain barrier is denied. Recently, we demonstrated that the GM1 oligosaccharide head group (GM1-OS) is the GM1 bioactive portion that, interacting with the TrkA-NGF complex at the membrane surface, promotes the activation of a multivariate network of intracellular events regulating neuronal differentiation, protection, and reparation. Here, we evaluated the GM1-OS neuroprotective potential against the Parkinson’s disease-linked neurotoxin MPTP, which destroys dopaminergic neurons by affecting mitochondrial bioenergetics and causing ROS overproduction. In dopaminergic and glutamatergic primary cultures, GM1-OS administration significantly increased neuronal survival, preserved neurite network, and reduced mitochondrial ROS production enhancing the mTOR/Akt/GSK3β pathway. These data highlight the neuroprotective efficacy of GM1-OS in parkinsonian models through the implementation of mitochondrial function and reduction in oxidative stress.

## 1. Introduction

Parkinson’s disease (PD) is a progressive neurodegenerative disorder characterized by the overtime loss of dopaminergic neurons in the substantia nigra pars compacta (SNpc), the accumulation of α-synuclein (αS)-containing aggregates, and dopamine (DA) deficit in the striatum [1]. Although the disease has been known for more than 200 years, most drugs just relieve symptoms temporarily to ameliorate patient’s quality of life, but a therapy that slows the course of the disease is not available yet [2].

More than 95% of PD forms are idiopathic and the etiopathogenetic causes of the disease are not known. PD is a multifactorial disease with aging, and mitochondrial and lysosomal failure recognized among the most accredited causes of the disease’s onset [3,4,5]. In addition, the imbalance of lipid metabolism is another factor linked to PD etiopathogenesis [6] and, in this context, GM1 ganglioside has gained significant attention.

GM1 is a sialic acid-containing glycosphingolipid, enriched in the brain and abundant in the neuronal plasma membrane. It is essential for neuronal homeostasis and has a bioactive function modulating multiple cellular processes comprising survival pathways, calcium signaling, neuronal firing, and mitochondrial activity [7,8,9,10]. Reduced levels of GM1 ganglioside and GM2/GD2 synthase, the enzymes responsible for GM1 synthesis, have been found in post-mortem brain tissues (SNpc and striatum) of PD patients, with respect to age-matched non-affected subjects [9,11,12,13,14]. These findings reveal the relationship between reduced GM1 levels and PD pathogenesis.

Accordingly, mice with subnormal GM1 content due to haploinsufficiency of GM2/GD2 synthase, spontaneously develop all the neurological features typical of human PD, demonstrating that deficient GM1 levels cause PD neurological defects [8,11,12,15,16].

Strategies aimed at increasing GM1 levels have been attempted as replacement therapy in PD. However, systemic administration of exogenous GM1 showed low efficacy due to the low penetrance of GM1 across the blood–brain barrier (BBB) [17,18,19].

In recent years, our group has discovered that the GM1 oligosaccharide (GM1-OS) constitutes the bioactive moiety of the ganglioside, and the biological properties are retained even when the saccharide is separated from the amphiphilic tail (ceramide). Specifically, interacting with the plasma membrane TrkA receptor, GM1-OS modulates neuronal differentiation, development, and protection from insults [20]. Importantly, in the GM1-deficient mouse model of PD, GM1-OS systemic administration led to DA neurons survival in SNpc, cleared αS aggregates, normalized striatal DA levels, and recovered motor symptoms [21].

1-methyl-4-phenyl-1,2,3,6-tetrahydropyridine hydrocholoride (MPTP) is a potent neuronal toxin that, when injected in humans and animals, induces parkinsonian symptoms [22,23]. Once in the blood stream, MPTP crosses the BBB and reaches the central nervous system (CNS). Here, astrocytes and other glial cells convert MPTP into the neurotoxic active metabolite 1-methyl-4-phenylpyridinium (MPP^+^). MPP^+^ enters DA neurons via the dopamine transporter (DAT) and inhibits the complex I of the respiratory chain, causing energy deficit and overload of reactive oxygen and nitrogen species, ultimately leading to neurodegeneration [24,25,26,27]. Defects in mitochondrial function have long been implicated in the onset of PD and wide evidence supports a central role of mitochondria alterations in PD progression [28,29,30]. GM1 has been reported to regulate mitochondrial activity and to protect DA neurons in MPTP models [31,32,33,34]. Notably, we found that GM1-OS protects neuroblastoma cells from MPTP damage and modulates mitochondria functions [32,35].

In this work, we aimed to study the potential protective effect of GM1-OS against MPTP in primary neurons. Our data indicate that GM1-OS sustained the survival of primary neurons by enhancing the mammalian target of rapamycin (mTOR)/RAC-alpha serine/threonine-protein kinase (Akt)/Glycogen synthase kinase-3β (GSK-3β) pathway and limiting the overload of mitochondrial reactive oxygen species (ROS).

## 2. Materials and Methods

### 2.1. Materials

Phosphate buffered saline (PBS), Calcium Magnesium Free (CMF)-PBS, glucose, RNAase-free water, paraformaldehyde (PFA), sodium orthovanadate (Na_3_VO_4_), phenylmethanesulfonylfluoride (PMSF), aprotinin, protease inhibitor cocktail (IP), horseradish peroxidase (HPR), o-phenylenediamine tablets, bovine serum albumin (BSA), cytarabine (Ara-c), ethylenediaminetetraacetic acid (EDTA), DNase I (used for neuronal culture), Trypsin, MPP^+^, sodium dodecyl sulfate (SDS), 2-propanol, formic acid, 3-(4,5,-dimethylthiazole-2yl)-2,5-diphenyletrazolium bromide (MTT), lactate dehydrogenase (LDH) release assay (MAK380), and DPX Mountant (06522) were from Sigma-Aldrich (St. Louis, MO, USA). Fetal Bovine Serum (FBS), Fetal Calf serum (FCS), L-Glutamine, penicillin/streptomycin (P/S) solution, and Hanks’ balanced salt solution (HBSS) were from EuroClone (Paignton, UK). MitoSOX™ Red superoxide indicator (M36008), Hoechst solution (33342), Neurobasal A medium, B27 Supplement and 4′,5-Diamidina-2-phenylindole (DAPI) were from Thermo Fischer Scientific (Waltham, MA, USA). 4–20% Mini-PROTEAN^®^ TGX™ Precast Protein Gels, Turbo Polyvinylidene difluoride (PVDF) Mini-Midi membrane and DC™ protein assay kit were from BioRad (Hercules, CA, USA). Leibovitz (L15) medium, Dulbecco’s modified Eagle’s medium (DMEM), Neurobasal medium, B27 Supplement, brain-derived neurotrophic factor (BDNF), and glial-derived neurotrophic factor (GDNF) were from Life Technologies (St. Christophe, France).

#### Antibodies

For immunofluorescence analyses of DA neurons, the following antibodies were used: primary mouse monoclonal anti-tyrosine hydroxylase (TH) antibody (RRID:AB_477560) and secondary goat anti-mouse IgG (H+L), coupled with an Alexa Fluor 488 antibody (RRID:AB_2532075) purchased from Sigma-Aldrich (St. Louis, MO, USA).

For western blotting (WB) analyses on cerebellar granule neurons (CGNs) lysates, the following antibodies were used: rabbit anti-mTOR (RRID:AB_330978), anti-phospho-mTOR (Ser2448, RRID AB_330970), rabbit anti-Akt (RRID:AB_329827), mouse anti-phospho-Akt (Ser473, RRID:AB_331158), rabbit anti-GSK-3β (RRID:AB_490890), anti-phospho-GSK-3β (Ser9, RRID:AB_2798546) primary antibodies, goat anti-rabbit IgG secondary antibody (RRID: AB_2099233) and goat anti-Mouse IgG (H+L) secondary antibody (RRID: AB_228307) were from Cell Signaling Technologies (Danvers, MA, USA).

### 2.2. Primary Cultures

#### 2.2.1. Primary Culture of DA Neurons

Rat DA neurons were cultured as previously described [24,25]. Briefly, 15-day-old pregnant female rats (Wistar, Janvier, France) were sacrificed using a deep anesthesia with CO_2_ chamber and a cervical dislocation. Fifteen-day-old rat embryos midbrain were dissected under a microscope. Embryonic midbrains were removed and placed in ice-cold L15 medium containing 2% P/S solution (penicillin 10,000 U/mL and streptomycin 10 mg/mL) and 1% of BSA. The ventral portion of the mesencephalic flexure, a region rich in DA neurons, was used for cell preparations. Middle brains were disassociated by trypsinization (20 min at 37 °C, 0.05% trypsin and 0.02% EDTA solution). The reaction was stopped by the addition of DMEM containing DNase I grade II (0.5 mg/mL) and FCS (10%). Then, cells were mechanically dissociated by three passes through a 10 mL pipette, centrifuged at 180× *g* (10 min, 4 °C) on a layer of BSA (3.5%) in L15 medium. The supernatant was discarded and the cell pellets were resuspended in a specific culture medium of Neurobasal supplemented with 2% B27, 2 mM L-glutamine, 2% P/S solution, 10 ng/mL BDNF, and 1 ng/mL GDNF, allowing to obtain a near pure neural cultures (<0.5% glia) with ~1–5% TH-positive cells on the total number of cells.

Viable cells were counted in a Neubauer cytometer using the trypan blue exclusion assay.

Cells were seeded at a density of 40,000 cells/well in 96-well plates (poly-L-lysine pre-coated) and maintained in a humidified incubator at 37 °C in 5% CO_2_/95% air atmosphere. Further, 24 h after seeding, in order to obtain enriched cultures of primary rat neuron, cytarabine was added to a final concentration of 7.5 μM to suppress glial proliferation. Every 2 days, half of the medium was changed to fresh medium.

#### 2.2.2. Primary Culture of CGNs

CGNs were prepared as previously described [36]. Briefly, 5-day-old C57BL/6J pups were sacrificed by decapitation to extract the cerebellum. CGNs were dissociated from cerebellar pool by mechanic trituration with blade (70 times in perpendicular directions), followed by incubation with 1% Trypsin and 0.1% DNase I in 0.2% CMF-PBS-glucose (1 mL/5 cerebella) for 3.5 min at 23 °C. The reaction was stopped by centrifugation of the cell mixture (1000× *g*, 10 s) and the supernatant was removed. The cell pellet was suspended in the 0.04% Trypsin inhibitor and 0.1% DNase I 0.1% in 0.2% CMF-PBS-glucose (1 mL/5 cerebella). The cells were finally dissociated by repeated passages in Pasteur glass of decreasing caliber. The reaction solution was removed by centrifugation (1000× *g*, 5 min) and cells were washed with 0.2% glucose in CMF-PBS before being resuspended in Neurobasal A medium containing 25 mM KCl, 1% B27 Supplement, 1% L-Glutamine and 1% P/S solution. Cells were counted using a Burker chamber, plated at a density of 315,000 cells/cm^2^ on plastic or glass coverslips, both precoated with poly-L-lysine (10 µg/mL for 2 h at 37 °C) and maintained at 37 °C in a humidified atmosphere of 95% air/5% CO_2_.

### 2.3. Cell Treatments

On day 14 of the culture, GM1-OS was dissolved in H_2_O at a stock concentration of 2 mM, further diluted in the culture medium at the final concentration of 100 μM [36]. Then, 1 h after GM1-OS incubation, MPP^+^ was added to a final concentration 4 µM (for DA neurons) or 50 µM (for CGNs) [25,37]. Control cells were incubated under the same experimental conditions but received vehicle (H_2_O) instead of GM1-OS and MPP^+^.

### 2.4. Cell Immunostaining

DA neurons were fixed by a solution of 4% PFA in PBS (pH = 7.3) for 20 min at 23 °C. They were washed twice again in PBS. The cell membranes were permeabilized and non-specific sites were blocked with a solution of PBS containing 0.1% of saponin and 1% of FCS, for 15 min at 23 °C. Cells were incubated with a mouse monoclonal anti-TH (1/1000 in PBS containing 1% FCS, 0.1% saponin) for 2 h at 23 °C. This antibody recognizes specifically DA neurons and neurites, allowing the study of their cell survival and neurite network [24]. This antibody was revealed with secondary antibody goat anti-mouse IgG coupled with an Alexa Fluor 488 (1/400 in PBS containing 1% FCS, 0.1% saponin), for 1 h at 23 °C.

For each condition, 20 pictures per well were automatically taken using ImageXpress (Molecular Devices) at 10X magnification. All images were generated using the same acquisition parameters. From images, analyses are directly and automatically performed by MetaXpress^®^ (Molecular Devices).

The following read-outs were investigated: (i) DA neuron survival (number of TH-positive neurons); (ii) and total neurite network of DA neurons (length of TH-positive neurites in µm).

### 2.5. MitoSOX Red Staining

After treatments, cultured CGNs were incubated with 1 μM MitoSOX™ Red reagent in HBSS with Ca^2+^ and Mg^2+^ for 10 min at 37 °C. At the end of the incubation, the cells were rinsed in PBS and fixed in 4% PFA (in PBS) for 20 min at 23 °C. Nuclei counterstaining was performed by Hoechst dye (0.0002% *v*/*v* in PBS, 5 min, 23 °C) and slides were mounted with DPX reagent. To assess the specific staining of mitochondrial O_2_^•−^, the morphology of stained intracellular structures was verified by analyzing CGNs with a NikonEclipse Ni upright microscope with a 100× objective (Appendix A). For the quantitative analyses, images were acquired using a 40× objective and the red signal was quantified with ImageJ software developed at the National Institutes of Health [38]. At least 10 fields for each condition were acquired for each experiment.

### 2.6. Morphological Analysis

Cultured CGNs, treated or not with GM1-OS, in the absence or presence of MPP^+^, were observed by phase contrast microscopy (20× magnification, Olimpus BX50 microscope, Olympus, Tokyo, Japan). At least 10 fields from each well were acquired for each experiment.

### 2.7. MTT Assay

Cell viability was determined by the MTT method in cultured CGNs, treated or not with GM1-OS, with or without of MPP^+^ [32,39]. Briefly, at the end of incubation, 2.4 mM MTT (4 mg/mL in PBS) was added to each well for 4 h at 37 °C. The medium was carefully removed and replaced with 2-propanol: formic acid, 95:5 (*v*/*v*). Plates were gently shaken prior to reading the absorbance at 570 nm with a microplate spectrophotometer (Wallac 1420 VICTOR2TM, Perkin Elmer, Waltham, MA, USA).

### 2.8. LDH Assay

The MPP^+^ cytotoxicity and GM1-OS cell protection were evaluated by an LDH release assay (MAK380) following manufacturer instructions (Sigma-Aldrich (St. Louis, MO, USA). Briefly, the cells plated into 96 mw were centrifuged at 600× *g* for 10 min to let cells precipitate, and 100 μL of cell supernatant was moved to a new 96 mw. A total of 100 μL of LDH Reaction Mix was added to each well, mixed, and incubated for 30 min at 23 °C. At the end of the incubation, 10 μL of Stop Solution was added to each well and the absorbance values at 450 nm were acquired with a microplate spectrophotometer (Wallac 1420 VICTOR2TM, Perkin Elmer, Waltham, MA, USA).

### 2.9. WB Analysis

CGNs incubated in the presence or not of MPP^+^ and GM1-OS were washed twice with cold PBS containing Na_3_VO_4_ (1 mM) and lysed with Na_3_VO_4_ (1 mM) PMSF (1 mM), aprotinin (2%), and IP (1%) in cold PBS. Proteins in equal amounts were denatured, run on 4–20% precast polyacrylamide gels by SDS-PAGE, and transferred onto PVDF membranes using the Trans-Blot^®^ Turbo™ Transfer System (Bio-Rad, Hercules, CA, USA).

The presence of mTOR, mTOR^P-Ser2448^, Akt, Akt^P-Ser473^, GSK-3β, and GSK-3β^P-Ser9^ was determined by specific primary antibodies (diluted 1/1000 in 5% BSA in TBS-0.1% tween), followed by reaction with secondary HPR-conjugated antibodies (diluted 1/2000 in 5% BSA in TBS-0.1% tween). The data acquisition and analysis were performed using Alliance Uvitec (Eppendorf, Hamburg, Germany).

### 2.10. GM1-OS Preparation

The GM1-OS was prepared by ozonolysis of GM1 followed by alkaline degradation as previously described [35,36,40]. Briefly, GM1 was dissolved in methanol and slowly saturated with ozone at 23 °C. Methanol was then evaporated under vacuum and the residue was immediately brought to pH = 10.5–11.0 by triethylamine addition. After 3 days, triethylamine was evaporated and GM1-OS was purified using chloroform:methanol:2-propanol:water (60:35:5:5 v:v) as an eluent by flash chromatography. The oligosaccharide was dissolved in methanol and stored at 4 °C. Altogether, nuclear magnetic resonance, mass spectrometry, and high performance thin layer chromatography analyses showed a homogeneity over 99% for the prepared oligosaccharide [21].

### 2.11. Protein Determination

Protein concentrations of samples were assessed using a DC™ protein assay kit according to the manufacturer’s instructions, using known concentrations of BSA as standards.

### 2.12. Statistical Analysis

All values are expressed as mean ± standard error of the mean (SEM). Statistical analysis was performed by one-way ANOVA, followed by a Tukey’s multiple comparisons test. *p* < 0.05 was considered significant.

## 3. Results

### 3.1. GM1-OS Protection from MPTP in DA Neurons

We previously demonstrated that the GM1-OS was able to reduce the oxidative stress and confer protection against MPTP neurotoxicity by activating the TrkA signaling at a plasma membrane level in mouse neuroblastoma cells [32]. Given the evidence of a potential role of GM1-OS in counteracting the MPTP-induced toxicity, we aimed to assess whether GM1-OS was directly capable of protecting the DA neurons, the primary MPTP-target, and the specific neuronal population affected in PD.

As expected, we found that MPP^+^ exposure significantly reduced DA neurons survival, assessed by a significant reduction in the number of TH-positive neurons and their neurite network (Figure 1a,b). Importantly, MPP^+^-injured DA neurons pre-treated with GM1-OS showed less loss of cellular TH signal (Figure 1a). Furthermore, the GM1-OS administration preserved the neurite network of MPP^+^-injured DA neurons (Figure 1c).

These experiments highlighted the GM1-OS protective role in DA neurons, the highest sensitive neuronal population to MPTP-induced parkinsonism.

### 3.2. GM1-OS Protection from MPTP in CGNs

PD manifestation begins with dopaminergic dysfunction in the nigrostriatal pathways and progresses to heterogeneous impairment due to diffused involvement of the cerebral cortex, and other tissutal districts such as the cerebellum [41,42,43]. Considering the evolution of PD with the involvement of cerebellar neurons, we decided to continue our studies on the effects, mechanisms, and processes underlying GM1-OS protection against MPTP toxicity in mouse CGN. This neuronal population is the most abundant in the whole brain and in the cerebellum [44,45], which makes these neurons ideal for further biochemical evaluations where a high number of cells is required.

First, as reported in Appendix A, we assessed the MPP^+^ concentration that induced neuronal toxicity in CGNs by exposing cells to incremental MPP^+^ concentrations. The MTT viability assay highlighted that the toxic effect of MPP^+^ was concentration- and time-dependent with an EC_50_ of 25 μM at 24 h and EC_50_ of 50 μM at 48 h (Appendix A). The treatment with GM1-OS significantly increased the neuronal survival when CGNs were injured with 25 μM, 50 μM, and 100 μM MPP^+^ for 24 h. However, when CGNs received MPP^+^ concentrations higher than 100 μM, GM1-OS treatment no longer showed any protective effect (Appendix A). Further morphological analysis showed that MPP^+^ at 25, 50, and 100 μM caused a progressive destruction of the CGNs neurite network, which was saved in the presence of GM1-OS (Appendix A). For subsequent evaluations, we chose the experimental paradigm using 50 μM MPP^+^ for 24 h.

By contrast phase imaging, we confirmed that MPP^+^-injured CGNs showed neurite fragmentation, which was preserved in the presence of GM1-OS (Figure 2a).

Moreover, GM1-OS improved the survival rate of intoxicated CGNs, and reduced the cell membrane permeability of MPP^+^-injured CGNs, as assessed by MTT and LDH assays, respectively (Figure 2b).

In order to evaluate the molecular factors modulated by GM1-OS in increasing neuronal survival, we focused on the mTOR/Akt/GSKβ pathway, which, in physiological conditions, sustains neuronal viability [46,47]. Specifically, mTOR activates Akt by phosphorylating the serine 473 residue (Ser473) through mTOR complex 2. Subsequently, P-Akt inhibits its downstream substrate GSK-3β, by phosphorylating the Ser9 residue. To note, GSK-3β signaling was previously reduced in DA neurons injured by MPTP and other insults [48,49]. Thus, we evaluated whether GM1-OS promoted neuronal survival by regulating the mTOR/Akt/GSK-3β signaling pathway. As shown in Figure 3, MPP^+^ administration decreased the phosphorylation level of both mTOR and Akt in CGNs. This effect was already observed after 6 h from MPP^+^ exposure. However, GM1-OS treatment alleviated the inhibitory effect on the mTOR and Akt phosphorylation caused by MPP^+^ and reversed the P-mTOR/mTOR and P-Akt/Akt ratio significantly. Similarly, MPP^+^ administration induced the activation of GSK3β, reducing the levels of phosphorylation at Ser9 residue, and GM1-OS treatment abolished the MPP^+^-induced activation of GSK-3β after 24 h (Figure 3). No significant changes in total mTOR, Akt and GSK-3β were observed after MPP^+^ and GM1-OS treatments, at any time point analyzed. Overall, these findings indicate that GM1-OS sustained neuronal survival by regulating the intracellular mTOR/Akt/GSK-3β signaling.

### 3.3. GM1-OS Lowered the Excess of Mitochondrial Superoxide (O_2_^•−^) Induced by MPTP

In target neurons, MPP^+^ affects mitochondrial function by inhibiting the respiratory complex I, disrupting the natural flow of electrons transport chain [30]. In this way, MPP^+^ causes an acute ATP deficiency and an increased leakage of ROS, particularly O_2_^•−^ from the respiratory chain [30].

We previously showed that GM1-OS was able to protect Neuro2a neuroblastoma cells from MPTP-induced mitochondrial oxidative stress by modulating mitochondrial biogenesis and functionality [32,35]. Considering that GM1-OS sustained the viability and neuronal integrity in DA and CG neurons injured with MPP^+^, we evaluated the levels of mitochondrial O_2_^•−^. Mitochondrial O_2_^•−^ was identified with the specific MitoSOX Red dye [32], after 1 h and 6 h of MPP^+^ exposure, two time points preceding the loss of cell viability (Figure 4). MPP^+^ significantly increased the O_2_^•−^ levels immediately after 1 h after administration. Importantly, GM1-OS treatment lowered mitochondrial O_2_^•−^ overload in MPP^+^ CGNs, at both analyzed time points (Figure 4). These results suggest that GM1-OS has neuroprotective properties against the parkinsonian pathogenesis induced by MPTP involved the modulation of mitochondrial activity.

## 4. Discussion

It was previously reported that methods to increase GM1 levels, either by the brain infusion of Vibrio Cholera sialidase or by exogenous GM1 injection, were effective in sparing DA neurons of the SNpc in MPTP mice [31].

Here, using the most sensitive neuronal population affected in PD we showed that GM1-OS protected primary mouse DA neurons exposed to the active metabolite of MPTP (MPP^+^). In fact, GM1-OS sustained the TH immunosignal, drastically reduced by the MPP^+^ toxin in DA neurons (Figure 1). Previous studies employing rat mesencephalic cultures injured with MPP^+^ found that exogenous GM1 partially rescued the levels of TH-positive neurons [50,51]. The protetive effect of GM1-OS sparing the TH signal in rat DA neurons is an additional in vitro demonstration of GM1-OS ability to retain the biological function of the parental ganglioside.

To further investigate the biochemical mechanisms underlying GM1-OS neuroprotection against MPP^+^, we used mouse CGN, another neuronal population implicated in PD [41,42,43,52]. Notably, to induce substantial damage in CGNs, higher MPP^+^ concentrations (25–100 μM) than those employed in DA neurons (4 μM) were required (Figure 1 and Appendix A). This indicates that DA neurons are more sensitive to MPP^+^ in line with the fact that they belong to the most affected neuronal populations in PD. As for DA neurons, GM1-OS increased the survival of CGNs and their neurite network (Figure 2).

It has been previously reported that the mTOR/Akt/Gsk3β pathway is critical for neuronal survival [46,47]. Exposure to MPP^+^ and other PD-related insults decreased the Akt phosphorylation and increased GSK3β activation leading to neurodegeneration. In MPP^+^-damaged CGNs, GM1-OS treatment alleviated the MPP^+^-induced changes in phosphorylated mTOR, Akt, and GSKβ levels indicating that mTOR/Akt/GSK3β can be a neuronal intracellular pathway modulated by GM1-OS (Figure 3). We previously reported that GM1-OS interacts with TrkA on the cell surface, forming a tripartite complex together with extracellular NGF, and activating an intracellular cascade of factors involving MAP kinases and PI3K [20]. Specifically, Akt and GSKβ are downstream of this pathway and the enhancement of their phosphorylation mediated by GM1-OS in the presence of MPP^+^, may likely be a consequence of activated neutrophin signaling at the neuronal plasma membrane.

Failure of mitochondrial function is another prominent feature characterizing the neurodegeneration accompanying aging and idiopathic PD, as demonstrated by the reduction in respiratory complex I in the brain of PD patients [53]. The in vivo and in vitro MPTP models recapitulate this pathogenetic aspect, where chemically induced mitochondrial deficiency is accompanied by redox imbalance with overproduction of reactive oxygen and nitrogen species, leading to neurodegeneration [30,54]. Our study demonstrates that GM1-OS prevented MPP^+^-induced mitochondrial O_2_^•−^ overload in primary neurons (Figure 4). This finding is consistent with the effect of GM1-OS in reducing mitochondrial ROS in MPTP-exposed neuroblastoma cells [32].

Althogh the MPTP in vitro and in vivo models do not fully recapitulate the multifactorality of PD (i.e., αS aggregation), they recapitulate the pro-inflammatory processes occuring in PD pathology [23]. In fact MPTP triggers inflammatory processes characterized by T-cell infiltration into the striatum and SNpc with microglia activation [30,54] and the release of pro-inflammarory factors (i.e., TNF-a, IL-6, and ROS) [55]. In the future we will evaluate whether GM1-OS is able to alleviate the inflammatory response of glial cells and confirm its potential neuroprotective role in MPTP mice.

## 5. Conclusions

The present study exploits the in vitro MPTP model of parkinsonism providing evidence of GM1-OS neuroprotective and neurorestorative properties of the damaged nigrostriatal DA system and glutamatergic neurons, trough the modulation of mTOR/Akt/GSK3β downstream pathway and reducing mitochondria ROS overproduction. Previously, numerous preclinical in vitro and in vivo studies highlighted the GM1 ganglioside ability to ameliorate PD associated MPTP-neurotoxic insults. The data reported here confirm that GM1 properties are due to its oligosaccharide portion.

The capability of GM1-OS to cross BBB more efficiently than GM1 ganglioside [56] and rescue parkinsonian defects in multiple models of the disease [21], strengthens a possible clinical application of GM1-OS as PD therapeutic and opens new perspectives over other neurodegenerative disorders (i.e., Huntington and Alzheimer), where the efficacy of GM1 replacement therapy has been observed [17,19,57].

## Figures and Tables

**Figure 1 biomedicines-11-01305-f001:**
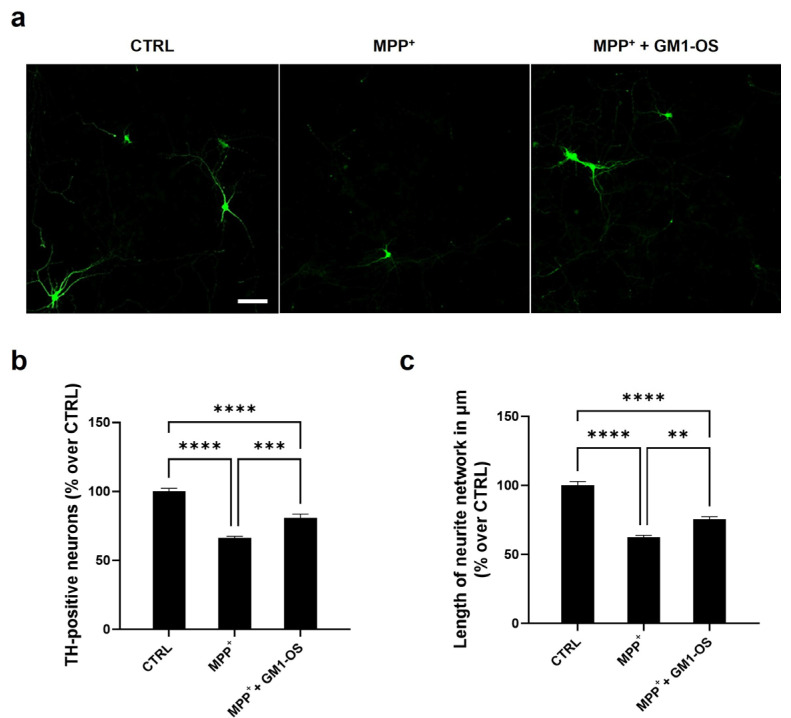
Neuroprotective effects of GM1-OS in primary cultures of rat DA neurons injured with MPP^+^. On day 6 of culture, primary DA neurons were pre-incubated with GM1-OS (100 μM) for 1 h, before MPP^+^ exposure. Next, MPP^+^ (4 µM) was added to the culture medium. After 48 h, TH immunofluorescence was performed as described in the methods section. (**a**) Representative immunofluorescence images of TH-positive neurons (10X magnifications); (**b**) number of TH-positive neurons expressed as fold change over untreated control (CTRL) cells, as read-out of DA neuron survival; (**c**) length of TH-positive neurites in µm, expressed as fold change over untreated CTRL cells, for neurite network of DA neurons. All values are expressed as mean ± SEM (n = 6, **** *p* < 0.0001; *** *p* < 0.001; ** *p* < 0.01; One-way ANOVA followed by Tukey’s multiple comparisons test).

**Figure 2 biomedicines-11-01305-f002:**
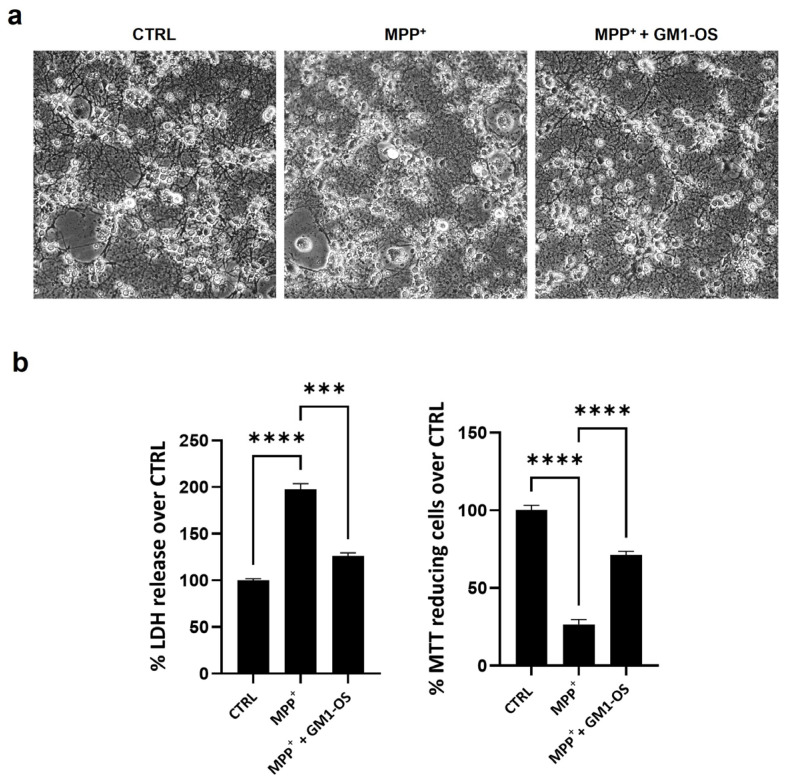
GM1-OS protected mouse CGNs injured with MPP^+^. On day 14 of culture, primary CGNs were pre-incubated with GM1-OS (100 μM) or water (CTRL) for 1 h before MPP^+^ exposure. Next, MPP^+^ (50 µM) or water was added to the culture medium for 24 h. (**a**) Phase contrast images of CGNs (20X magnification). Images are representative of four independent experiments (n = 4); (**b**) Viability assays: on the left LDH release, on the right MTT assay. All values are expressed as mean ± SEM (n = 4 independent experiments. *** *p* = 0.0005, **** *p* < 0.0001, one-way ANOVA followed by Tukey’s multiple comparisons test).

**Figure 3 biomedicines-11-01305-f003:**
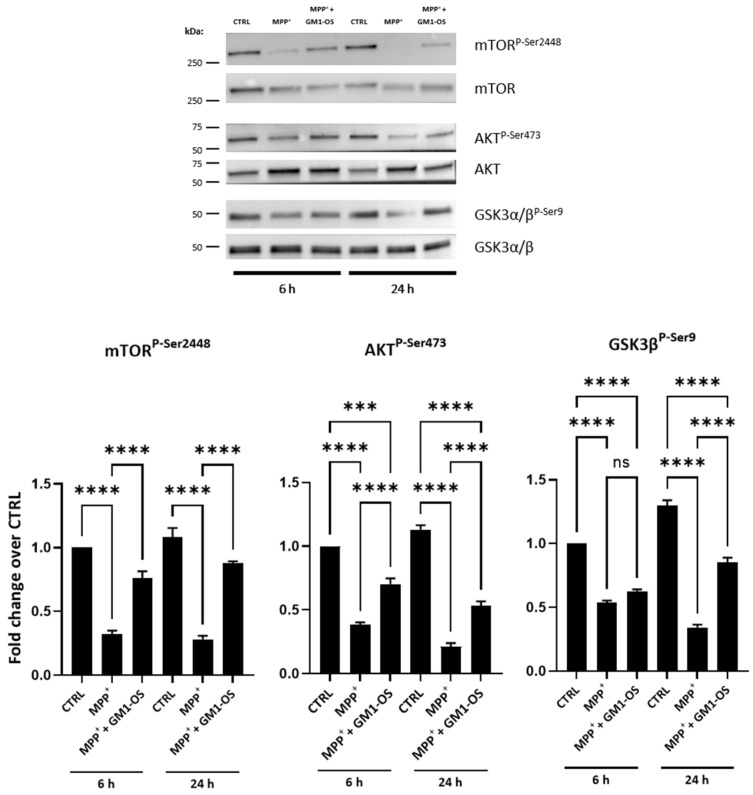
GM1-OS modulation of mTOR/Akt/GSK-3β pathway. WB of factors sustaining cell survival after 6 h and 24 h of MPP^+^ exposure: mTOR^P-Ser2448^, AKT^P-Ser473^, GSK3β^P-Ser9^, and respective total proteins. Top: WB representative images; bottom: semiquantitative analysis of WB bands represented as fold change over CTRL. All values are expressed as mean ± SEM (n = 4 independent experiments. *** *p* = 0.0002, **** *p* < 0.0001, one-way ANOVA followed by Tukey’s multiple comparisons test).

**Figure 4 biomedicines-11-01305-f004:**
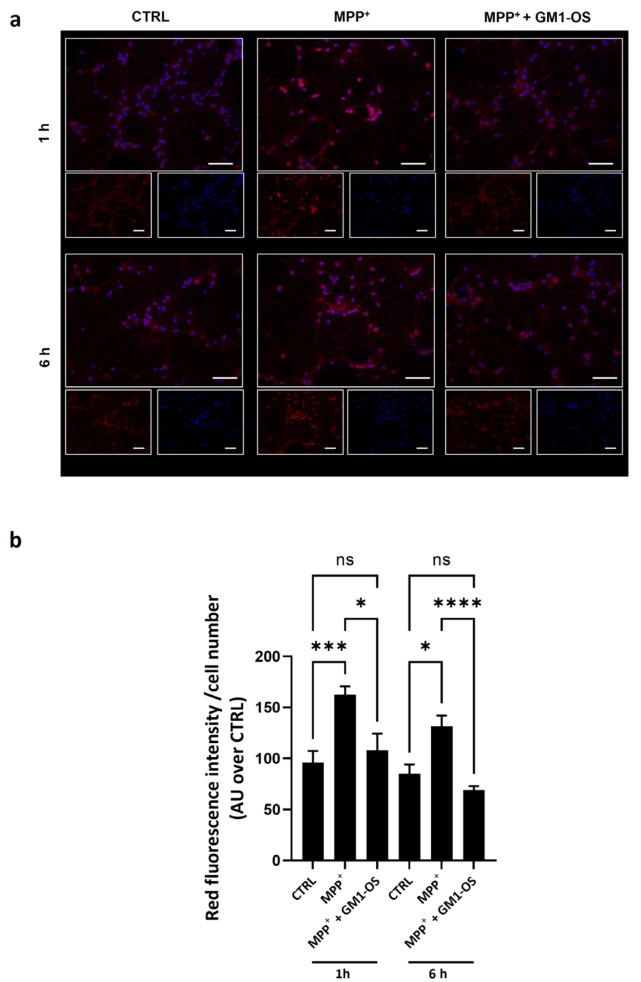
GM1-OS counteracted mitochondrial O_2_^•−^ increase induced by MPP^+^. On day 14 of culture, primary CGNs were pre-incubated with GM1-OS (100 μM) or water (CTRL) for 1 h before MPP^+^ exposure. Next, MPP^+^ (50 µM) or water was added to the culture and mitochondrial O_2_^•−^ was evaluated by MitoSOX Red reagent. (**a**) Representative fluorescence images of CGNs after 1 h (top) or 6 h (bottom) of MPP^+^ administration (40× magnification). For each condition the small quadrants show single channel images with MitoSOX in red and Nuclei in blue and the big quadrant is the overlayed image (Scale bar: 50 µm); (**b**) quantification of the MitoSOX Red signal over nuclei number. All values are expressed as mean ± SEM (n = 3 independent experiments. * *p* < 0.01, *** *p* = 0.0006 **** *p* < 0.0001, one-way ANOVA followed by Tukey’s multiple comparisons test).

## Data Availability

The data presented in this study are available upon reasonable request to the corresponding authors.

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
