# Peer review of "GM1 Oligosaccharide Efficacy in Parkinson’s Disease: Protection against MPTP"

_biomedicines, 2023, doi:10.3390/biomedicines11051305_

Round 1

Reviewer 1 Report

I have no serious remarks about the work.

1) It would be good to provide double staining of culture cells with GFAP and NeuN to see the ratio of neurons and astrocytes.

2) It is necessary to confirm the localization of MitoSox Red in mitochondria. I would recommend authors to load the cells at the same time MitoSox Red and Mito Tracker

3) The conclusion should not contain arguments, discussions and limitations of the results obtained. The authors need to strictly draw a conclusion on the basis of their own results.

Author Response

Revision according to the comments of Reviewer #1:

1) It would be good to provide double staining of culture cells with GFAP and NeuN to see the ratio of neurons and astrocytes.

Dopaminergic cultures were prepared accordingly to Visanji et al. [FASEB J. 2008, 22:2488-97] starting from 15-days-old rat embryos. Specifically, to suppress glial proliferation and obtain enriched cultures of primary rat primary, 24 h after seeding the cells, cytarabine was added to a final concentration of 7.5 μM as reported in Methods paragraph “2.2.1. Primary culture of DA neurons”, lines 150-151.

As previously demonstrated, we have a near pure neuronal culture (< 0.5% glia) that contained tyrosine hydroxylase-positive cells (1-5 % of the total number of cells in these cultures). This was achieved by maintain cells in serum-free Neurobasal medium supplemented with B27 (2 %), L-glutamine (2 mM) and 2 % of P/S solution and 10 ng/mL of BDNF and 1 ng/mL of GDNF.

We also have a well-established protocol to prepare co-cultures of dopaminergic neurons and glia cells. In this case, according to Zhang et al. 2005 [FASEB J 2005, 19:533-42], cultures are prepared starting from 14-days-old rat embryos. Here, cultures are maintained in a defined culture medium consisting of Neurobasal medium with a 2 % solution of B27 supplement, 2 mM L-glutamine, 2 % P/S, 10 ng/mL BDNF, 1 ng/mL GDNF, 4 % heat-inactivated FCS, 1 g/L glucose, 1 mM sodium pyruvate, and 100 µM of non-essential amino acids. Any addition of cytarabine is avoided to guaranty the survival of proliferating cells (i.e. glia).

Accordingly, we added this sentence at lines 143-144: “allowing to obtain a near pure neural cultures (<0.5% glia) with ~1-5% TH-positive cells on the total number of cells”.

The subsequent analyses were conducted only on TH-positive cells.

2) It is necessary to confirm the localization of MitoSox Red in mitochondria. I would recommend authors to load the cells at the same time MitoSox Red and Mito Tracker.

In accordance with MitoSOX Red datasheet (Thermo Fischer Scientific), this dye is specifically designed to selectively target mitochondria. Accordingly, in literature several studies load cells only with MitoSOX red in order to stain mitochondrial ROS. For examples:

  1. Wang Y, Guan X, Gao CL, Ruan W, Zhao S, Kai G, Li F, Pang T. Medioresinol as a novel PGC-1α activator prevents pyroptosis of endothelial cells in ischemic stroke through PPARα-GOT1 axis. Pharmacol Res. 2021 Jul;169:105640. doi: 10.1016/j.phrs.2021.105640. Epub 2021 Apr 27. PMID: 33915296.
  2. Zhao P, Wei Y, Sun G, Xu L, Wang T, Tian Y, Chao H, Tu Y, Ji J. Fetuin-A alleviates neuroinflammation against traumatic brain injury-induced microglial necroptosis by regulating Nrf-2/HO-1 pathway. J Neuroinflammation. 2022 Nov 4;19(1):269. doi: 10.1186/s12974-022-02633-5. PMID: 36333786; PMCID: PMC9636801.
  3. Yazdankhah M, Ghosh S, Shang P, Stepicheva N, Hose S, Liu H, Chamling X, Tian S, Sullivan MLG, Calderon MJ, Fitting CS, Weiss J, Jayagopal A, Handa JT, Sahel JA, Zigler JS Jr, Kinchington PR, Zack DJ, Sinha D. BNIP3L-mediated mitophagy is required for mitochondrial remodeling during the differentiation of optic nerve oligodendrocytes. Autophagy. 2021 Oct;17(10):3140-3159. doi: 10.1080/15548627.2020.1871204. Epub 2021 Jan 19. PMID: 33404293; PMCID: PMC8526037.
  4. He Y, Zheng Z, Liu C, Li W, Zhao L, Nie G, Li H. Inhibiting DNA methylation alleviates cisplatin-induced hearing loss by decreasing oxidative stress-induced mitochondria-dependent apoptosis viathe LRP1-PI3K/AKT pathway. Acta Pharm Sin B. 2022 Mar;12(3):1305-1321. doi: 10.1016/j.apsb.2021.11.002. Epub 2021 Nov 9. PMID: 35530135; PMCID: PMC9069410.
  5. Zhao X, Wang C, Dai S, Liu Y, Zhang F, Peng C, Li Y. Quercetin Protects Ethanol-Induced Hepatocyte Pyroptosis via Scavenging Mitochondrial ROS and Promoting PGC-1α-Regulated Mitochondrial Homeostasis in L02 Cells. Oxid Med Cell Longev. 2022 Jul 16;2022:4591134. doi: 10.1155/2022/4591134. PMID: 35879991; PMCID: PMC9308520.

However, we agree that excessive overload of cells can lead an improper localization of the dye. For this reason, we have set up the staining protocol in order to use a low concentration of dye corresponding to 1 μM instead of 5 μM that is suggested by manufacturer.

In order to appreciate mitochondrial localization of MitoSOX Red dye in mitochondria, we now provide, as a new Supplementary Figure S1, a representative image of the staining acquired with Nikon Eclipse Ni upright microscope with 100X objective.

We added this sentence to Methods paragraph “2.5. MitoSOX Red staining”, lines 205-208: “To assess the specific staining of mitochondrial O2•-, the morphology of stained intracellular structures was verified by analyzing CGN with a Nikon Eclipse Ni upright microscope with a 100X objective (Supplementary Figure S1)”.

3) The conclusion should not contain arguments, discussions and limitations of the results obtained. The authors need to strictly draw a conclusion on the basis of their own results.

We modified the Conclusion section following the Reviewer's suggestions.

Reviewer 2 Report

In the present study, the authors used CGN instead of DA neurons because a high number of cells was required for biochemical evaluations. The mechanisms of MPTP-induced neuronal toxicity and GM1-OS protection were same between DA neurons and CGN, or not?

In Figure 3, the bands of mTOR were present on just below the lower limit of membranes. The images that clearly show the entire bands should be presented. In addition, was the expression of Akt (not phosphorylated) increased by MPP+?

In Figure 4a, scale bar was lost. There were three images in each group. The explanation of each image, MitoSOX Red, nuclei, and merged, should be added in figure legends.

There were some typographical errors; such as "Length" in Figure 1c, "contrast" in line 303, and so on.

Author Response

Revision according to the comments of Reviewer #2:

  • In the present study, the authors used CGN instead of DA neurons because a high number of cells was required for biochemical evaluations. The mechanisms of MPTP-induced neuronal toxicity and GM1-OS protection were same between DA neurons and CGN, or not?

Yes, the MPTP-induced neuronal toxicity and GM1-OS protection were same between DA neurons and CGNs. As shown in Figure 1 and Figure 2. as far as we could test, both MPP+ exposed DA neurons and CGNs lost neurite network integrity and this effect was prevented by the GM1-OS treatment (Figure 1: TH ICC in DA neurons; Figure 2a and Figure S2b: phase contrast images of CGNs). As we mentioned in the Discussion section, the two neuronal populations differed for their level of sensitivity to MPP+. Indeed, DA neurons showed signs of neurite disruption when incubated with 5 μM MPP+, while CGNs were more resilient, since they started to manifest signs of cell stress with [MPP+] > 25 μM for 24 and 48 h. No obvious sign of CGNs stress was observed for [MPP+] < 25 μM at the same time points.

To specifically verify the molecular mechanism underling the GM1-OS effect we used only CGN cells due to the material availability for biochemical investigations. As we reported at lines 335-354, to evaluate the molecular mechanisms underlying MPP+ injury in neurons, we specifically focused on mTOT/Akt/GSK-3β pathway that has been previously reported to be affected in DA neurons, challenged with MPP+ [47,48]. We showed that the same mechanism is affected by MPP+ in CGNs, and GM1-OS sustained cell viability and neurite integrity by modulating that pathway.

  • In Figure 3, the bands of mTOR were present on just below the lower limit of membranes. The images that clearly show the entire bands should be presented. In addition, was the expression of Akt (not phosphorylated) increased by MPP+?

We agree with the Reviewer comment about the quality of P-mTOR and total mTOR WB bands shown and we made a modification to the Figure 3 with another pair of representative images.

Regarding the total levels of mTOR, Akt and GSK-3β, our results show that they did not change by the MPP+ challenge and we clarified it in the main text by adding the following sentence at lines 349-351: “No significant changes of total mTOR, Akt and GSK-3β were observed after MPP+ and GM1-OS treatments, at any time point analyzed”.

  • In Figure 4a, scale bar was lost. There were three images in each group. The explanation of each image, MitoSOX Red, nuclei, and merged, should be added in figure legends.

We agree with the Reviewer comment and the scale bars were added to the ICC images in Figure 4.

The Figure 4 legend was modified as follows: “(a) Representative fluorescence images of CGN after 1 h (top) or 6 h (bottom) of MPP+ administration (40X magnification). For each condition, the small quadrants show single channel images with MitoSOX in red and Nuclei in blue and the big quadrant is the overlayed image. Scale bar: 50 µm”.

  • There were some typographical errors; such as "Length" in Figure 1c, "contrast" in line 303, and so on.

Typos have been corrected throughout the text.

Round 2

Reviewer 1 Report

The authors took into account all my comments. The article can be accepted for publication